# Exercise-Associated Hyponatremia During a Self-Paced Marathon Attempt in a 15-Year-Old Male Teenager

**DOI:** 10.3390/medicina55030063

**Published:** 2019-03-07

**Authors:** Beat Knechtle, Jonah Bamert, Thomas Rosemann, Pantelis T. Nikolaidis

**Affiliations:** 1Medbase St. Gallen Am Vadianplatz, 9000 St. Gallen, Switzerland; 2Institute of Primary Care, University of Zurich, 8006 Zurich, Switzerland; jonah.bamert@gmx.ch (J.B.); thomas.rosemann@usz.ch (T.R.); 3Laboratory of Exercise Testing, Hellenic Air Force Academy, 13671 Dekelia, Greece; pademil@hotmail.com

**Keywords:** adolescence, blood physiology, caloric intake, endurance, exercise-associated hyponatremia, fatigue, fluid intake, performance

## Abstract

*Background and objective*: The increased participation in endurance sports such as marathon running has attracted scientific interest especially with regard to adult athletes. However, few studies have examined the impact of a marathon race on children and adolescents. Therefore, the aim of the present case study was two-fold: first, to describe pacing during a marathon race, and second, to examine acute responses of blood physiology and biochemistry parameters during the race (i.e., pre- and post-race) as well as five consecutive days after the race. *Materials and Methods*: Participant was a 15-year-old boy who completed a self-paced marathon attempt for the first time and finished in 5 h 19 m 53 s. Positive pacing (i.e., a running speed that decreased throughout race) with a final end spurt was observed. *Results*: An increase in fluid intake across race was shown. Exercise-associated hyponatremia (EAH, i.e., plasma sodium concentration <135 mmol/L) was found post-race. C-reactive protein (CRP) did not correlate either with creatine kinase (CK) (r = 0.457, *p* = 0.302) or with lactate dehydrogenase (LDH) (r = 0.156, *p* = 0.739); however, leukocytes correlated very largely with LDH (r = 0.889, *p* = 0.007) but not with CK (r = 0.696, *p* = 0.082). CK and LDH related almost perfectly with creatinine (r = 0.937, *p* = 0.002 and r = 0.959, *p* = 0.001, respectively); also, creatinine clearance correlated very largely with CK (r = −0.782, *p* = 0.038) but not with LDH (r = −0.733, *p* = 0.061). Leukocytes, aspartate aminotransferase, LDH, and CK deviated from physiological range post-race, but returned to normal values during the five-day recovery period. *Conclusions*: In summary, a male teenager at the age of 15 years was able to run a marathon in under 6 h without significant harmful effects on health. He developed mild and asymptomatic EAH and an increase in leucocytes, CRP, CK, and LDH as markers of inflammation and skeletal muscle damage. EAH after the marathon was resolved within one day of recovery.

## 1. Introduction

Marathon running has predominantly been a domain of elite runners [1]; however, marathons recently have become attractive to both recreational and professional athletes [2]. In marathon running, pacing during the race is of high importance for a successful finish [3]. Although elite marathoners try to achieve a positive pacing with minimal speed change [1,2], recreational age group marathoners largely show a positive pacing with a continuous decrease during running as well as a final end spurt [3,4,5]. Although females and males 5–91 years old have demonstrated the ability run a half-marathon [6] and, 5–93 years old, a full marathon [7,8], little is known about the pacing of marathoners younger than 18 years because the official age limit to compete in a city marathon is set at 18 years [9].

Several studies have demonstrated that marathon running below the official age of 18 years seems not to be harmful in terms of health [10,11]. However, we have little knowledge about the pacing strategy during a self-paced marathon attempt or the recovery phase in a marathoner younger than 18 years [9]. In this case study, we investigated the changes in running speed during a marathon run and the changes in selected biochemical variables during the recovery phase in a 15-year-old teenager in a self-paced marathon attempt.

## 2. Experimental Section

### 2.1. Subject and Event

This case study was approved by the Institutional Review Board of EKOS (Ethikkommission Ostschweiz), Switzerland (EKOS 19/178). The runner and his parents provided their informed written consent for the case study. Our athlete is a 15-year-old boy (173 cm, 55 kg) born 19 July 2003. On 10 November 2018, he performed a self-paced marathon attempt at the age of 15 years, 3 months, and 22 days. The athlete prepared for this event using the recommendations of his parents, his grandparents, and his teacher regarding training and equipment. No specific recommendations were followed regarding food and fluid intake before, during, and after the event. He trained for his attempt by using the recommendations of a former elite Swiss marathoner (www.vikmotion.ch/wp-content/uploads/2016/06/VikMotion_Trainingsplan_Mara- thon.pdf). The marathon course was a long loop from Bad Ragaz to Buchs and back along the Rhine River, which forms the border between Switzerland and Austria in the eastern part of Switzerland.

On the day of the event, the weather was clear and sunny, with no precipitation. During the first half of the run (i.e., Bad Ragaz—Buchs), the runner faced a tailwind; in the second half (i.e., Buchs—Bad Ragaz), he faced a strong headwind. The temperature rose from 12.5 °C at the start to 19 °C by the end of the run. During the run, the support crew provided food (e.g., carbohydrate gels) and fluid (e.g., water) and recorded all details. In the first half of the run (i.e., Bad Ragaz—Buchs), two people acted as supporters. In the second half of the run (i.e., Buchs—Bad Ragaz), two additional people followed the athlete. The runner received liquid (i.e., pure water) and food (i.e., five carbohydrate gels and one energy bar) not according to a fixed scheme but according to his needs. Fluid intake was measured in deciliters; calorie intake was estimated using the consumer information on the product description. Sodium intake from the energy gels was 1010 mg and 195 mg from the energy bar. The athlete showed no signs of dehydration and/or hyponatremia.

### 2.2. Measurements

During the run, the heart rate of the athlete was continuously measured using POLAR M400 (Polar Electro Oy, Kempele, Finnland). After each kilometer, the heart rate of the athlete was recorded. The day before, after the run, and then for five days during the recovery, venous blood samples were drawn for hematological analysis, including erythrocytes, hemoglobin, hematocrit, thrombocytes, mean cell volume (MCV), mean cell hemoglobin (MCH), mean cell hemoglobin concentration (MCHC), and leucocytes. Biochemical analyses were also conducted with regard to C-reactive protein (CRP), creatine kinase (CK), lactate dehydrogenase (LDH), glutamate pyruvate transaminase (GPT), aspartate aminotransferase (GOT), gamma-glutamyl transferase (γ-GT), creatinine, and sodium. All samples were drawn by a professional nurse and transferred for analyses to a professional laboratory of Labormedizinisches Zentrum Dr. Risch (www.risch.ch). Creatinine clearance was estimated using the Cockcroft and Gault formula [12].

### 2.3. Data Analysis

The variation of speed and heart rate (HR) by kilometer and split was examined using a linear or a fourth-degree polynomial regression analysis, depending on which one polynomial provided the best fit according to F test; the relationship between these variables was estimated by coefficient of determination (R^2^). The distance was divided into six 7 km splits, i.e., 1–7 km, 8–14 km, 15–21 km, 22–28 km, 29–35 km, and 36–42 km. For the purpose of this analysis, a split refers to distance interval rather than to a specific distance point. A repeated measures analysis of variance examined differences in speed and HR among splits. Confidence intervals (CI) of 95% were calculated for mean differences among splits. The magnitude of differences was estimated using an eta-square classified as small (0.010 < η^2^ ≤ 0.059), medium (0.059 < η^2^ ≤ 0.138), and large (η^2^ > 0.138). The abovementioned statistics should be considered with caution in terms of the interpretation of the findings, keeping in mind the general nature of the study (case study, *n* = 1).

## 3. Results

The runner completed the full marathon distance in 5 h 19 m 53 s. During the run, fluid intake continuously increased, whereas caloric intake remained unchanged over time (Figure 1). Overall, the athlete drank 1.9 L during the run.

As shown in Figure 2, the pacing followed a positive pattern, i.e., the speed decreased across the race, and an endspurt was observed. A large main effect of a 7 km split was observed on speed (F_5_ = 3.169, *p* = 0.021, η^2^ = 0.346), where the last split was slower than the first by 2.2 km/h (95% CI; 0.1, 4.3). No main effect of the 7 km split on HR was shown (F_5_ = 0.756, *p* = 0.589, η^2^ = 0.112). Speed (r = 0.92, *p* = 0.010) and HR correlated perfectly or very largely (r = 0.87, *p* = 0.024) with distance (7 km splits).

The indices of blood physiology during the race and recovery are presented in Figure 3. The indices of biochemistry are presented in Figure 4. CRP did not correlate either with CK (r = 0.457, *p* = 0.302) or with LDH (r = 0.156, *p* = 0.739); however, leukocytes correlated very largely with LDH (r = 0.889, *p* = 0.007) but not with CK (r = 0.696, *p* = 0.082). CK and LDH related almost perfectly with creatinine (r = 0.937, *p* = 0.002 and r = 0.959, *p* = 0.001, respectively); also, creatinine clearance correlated very largely with CK (r = −0.782, *p* = 0.038) but not with LDH (r = −0.733, *p* = 0.061).

## 4. Discussion

It might be assumed that running a marathon at this age could be a health risk. However, several studies have found that running a marathon under the official age of 18 years poses no significant health risks [10,11]. During this self-paced marathon running attempt, where a male 15-year-old teenager finished the full marathon distance under 6 h, the official time limit for a marathon, we found (i) a positive pacing with a final end spurt, (ii) an increase in fluid intake during the run, and (iii) mild and asymptomatic exercise-associated hyponatremia (i.e., plasma sodium concentration <135 mmol/L) after the run.

### 4.1. Positive Pacing with a Final End Spurt

The runner showed a positive pacing with a final end spurt. A decrease in running speed with a final end spurt is very common in marathon running. During the 2015 New York City Marathon, in 20,283 women and 28,282 men competed, women and men of all age groups exhibited a reduced running speed over the course of the marathon with a final spurt in the last segment [13]. However, a decrease in running speed during a marathon might be a specific feature of a young athlete. It has been demonstrated that, for marathoners with a relatively slow overall pace, older athletes presented a relatively more even pacing in comparison with younger athletes [3]. A more even pacing is also a trait specific to male marathoners with a higher aerobic capacity and lower muscle strength [5]. Marathoners often try to maintain an even pace profile along the marathon course, partly by avoiding an excessively fast start that might result in a pronounced decrease in speed during the second half of the race [2].

### 4.2. Exercise-Associated Hyponatremia after the Run

The runner developed mild and asymptomatic EAH after the run with a plasma sodium concentration of 134 mmol/L. EAH is the most commonly reported electrolyte disorder associated with endurance performance [14,15,16,17,18] but does not always occur in a marathon run [19]. Several risk factors have been linked with EAH. Main risk factors include excessive consumption of water, sports drinks, and other hypotonic beverages; weight gain during exercise; exercise duration longer than four hours; event inexperience or inadequate training; slow running or performance pace; high or low body mass index; and limited availability of fluids [14,17,18,20,21]. However, although this 15-year-old teenager was inexperienced, age was not a risk in the development of EAH, especially when the marathon was finished within the time limit of 6 h. Forty-seven runners of 13–17 years completed a marathon with a mean finishing time of 4 h 57 min (ranging from 3 h 17 min to 6 h 14 min) and no case of EAH was found [11].

As time passed during the run, the athlete consumed more fluids but not more calories. Overall fluid intake during the run was 1.8 L. Fluid intake has been shown to be related to post-race plasma sodium levels. In marathoners competing in the 2006 Zürich Marathon, fluid intake correlated significantly and negatively with a change in plasma sodium between the start and finish of the race [15]. It has been shown that lighter and slower marathoners have a more positive fluid balance [17]. However, to develop severe EAH, fluid intake must be more than 1.8 L during a marathon. In marathoners competing in the 2002 Boston Marathon, hyponatremia was associated with consumption of more than 3 L of fluids during the race [16]. In official marathon races, slower runners consume more liquids than faster runners. To prevent overdrinking, no more than 0.4–0.8 L/h should be consumed [21]. Accordingly, our runner, who consumed 1.8 L within 6 h, was not at risk of overdrinking.

### 4.3. Changes in Laboratory Values

After the run, we found elevated values for leukocytes, GOT, LDH, CK, and CRP. By the end of the five-day recovery period, all values turned into reference range. This confirms recent findings in a 17-year-old female teenager in a self-paced marathon attempt [9]. An increase in CK as a sign of skeletal muscle damage [22,23] and an increase in leukocytes and CRP as parameters of an inflammation [24] are common findings in marathoners. We also found that creatinine and creatinine clearance increased after the run. However, both variables remained within the reference values. A marathon run may also lead to renal function abnormalities [25], including an increase in parameters of renal insufficiency, such as serum urea and serum creatinine, as well as a decrease in glomerular filtration rate [26]. These changes are attributable to various causes, such as dehydration or rhabdomyolysis [26].

One limitation of this case report was the inability to measure body mass and urinary parameters (e.g., urine specific gravity, urine osmolality) to quantify hydration status and signs of dehydration. Weight loss and changes in hydration status might have had an effect on the biomarkers.

In summary, in our study, a male teenager at the age of 15 years was able to run a marathon in under 6 h without significant harmful effects on health. He developed mild and asymptomatic EAH and an increase in leucocytes, CRP, CK, and LDH as markers of inflammation and skeletal muscle damage. After the marathon, EAH was resolved within one day of recovery. Given the lack of existing research and in light of the increased popularity of marathon running, this finding will be of practical value with regard to the acute physiological responses to a marathon race in adolescents. For instance, this case study can be used as an example in clinical practice when practitioners are questioned by the parents of young endurance runners about the safety of participating in a marathon race.

## Figures and Tables

**Figure 1 medicina-55-00063-f001:**
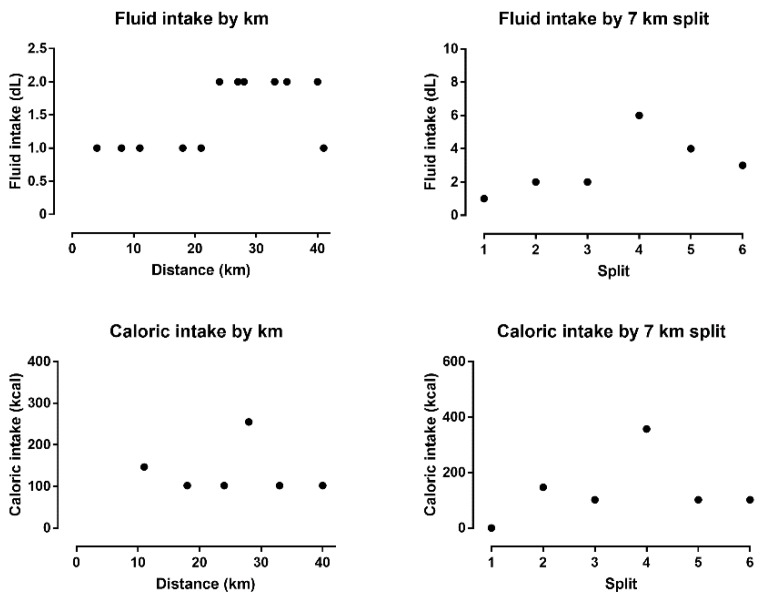
Variation of fluid and caloric intake by distance and split.

**Figure 2 medicina-55-00063-f002:**
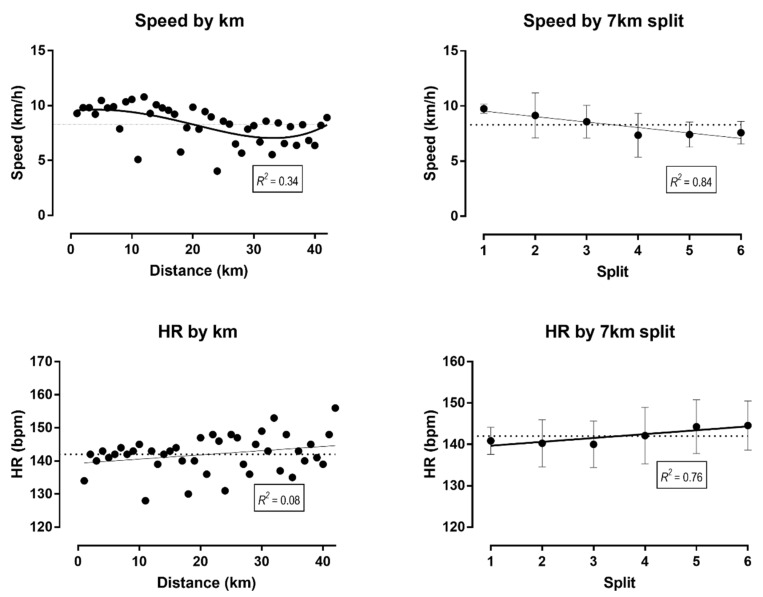
Variation of speed and heart rate by distance and split. *R*^2^ = coefficient of determination; HR = heart rate.

**Figure 3 medicina-55-00063-f003:**
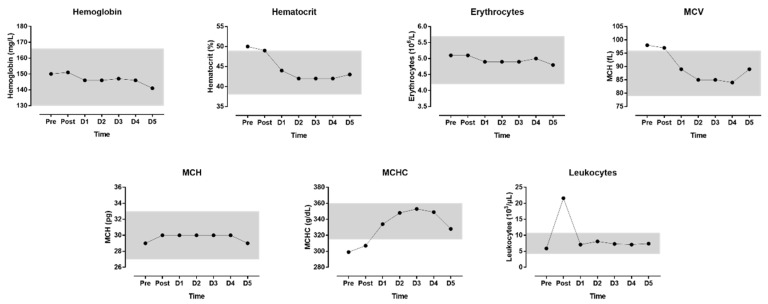
Blood physiology indices during the race and recovery. The shadowed area represents normal reference values (reference values from www.risch.ch/de/ribook). D = day post-race.

**Figure 4 medicina-55-00063-f004:**
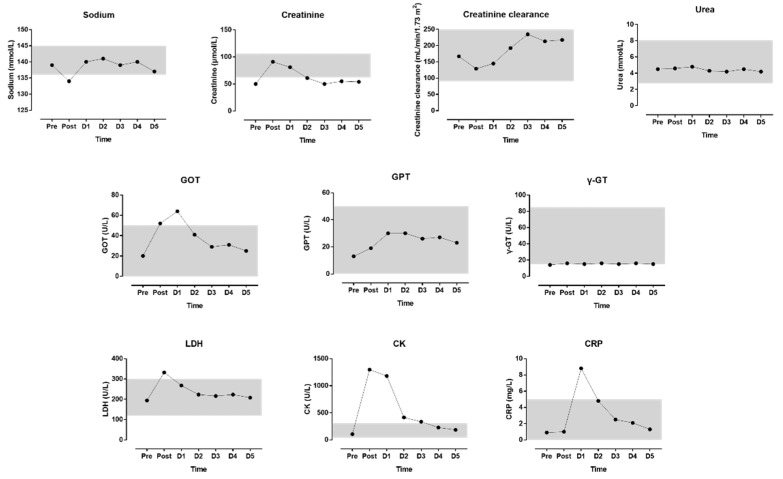
Biochemical indices during the race and recovery. The shadowed area represents normal reference values (reference values from www.risch.ch/de/ribook). D = day post-race.

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
