# Peer review of "Exercise-Associated Hyponatremia During a Self-Paced Marathon Attempt in a 15-Year-Old Male Teenager"

_medicina, 2019, doi:10.3390/medicina55030063_

Round 1
Reviewer 1 Report
Knechtle et al present an interesting case-study in an understudied population. While the data presented is certainly informative from a self-pacing point of view, I have some concerns about the interpretation and main findings of the paper.
LINE 51: What were the post-race instructions for the athlete? Was there any specific dietary or activity instructions? The 5-day post-race timepoint also has a fairly low sodium level, similar to the post-race value.
LINE 56: More details about the preparation are required. Was this self-taught? Was there a coach? Was there specific instructions for fluid intake and pacing in the pre-race recommendations? What was the pre-race preparation (fluids/food)? All these elements could impact both the pacing, and fluid intake findings of the study.
LINE 64: One of the main results of the study is that fluid intake increased in the second half of the run. Was this confounded by the additional 2 support team members that followed the athlete? Were fluids offered to the athlete, or asked for unprompted? If this is a major finding, more methodological details should be presented.
Some of the statistics seem inappropriate for the case study design. Since this is an n=1 study, I fear there is too much overinterpretation of the statistics presented for the data. Presentation of case data with subjective trend analysis is likely adequate for this paper; there isn’t enough to make use of more complicated statistics to interpret the findings for generalizability.
LINE 79: Why either linear or 4th degree polynomial? Was any F testing done to examine which polynomial gave the best fit?
LINE 79: Regressions with time-series data (or distance in this case – similar issue), do not give true relationships amongst variables, since the dependent is usually a function of time. The trend line is fine, but there is likely no need to show the confidence bars in the plots.
LINE 82: How was an ANOVA used when there was no variance at any of the Distance measures? It appears the authors grouped their km data into bins. Why 7km bins? 6 splits of 7km is 35km… where did the rest of the km get split?
FIG1: Fluid intake seems to be measured in an ordinal scale (either 1 or 2 dL units). It would be inappropriate to use any regression slopes with this data.
FIG4 & 5: Please provide a citation for the reference ranges.
LINE 122: Are the reference values for EAH appropriate for adolescent athletes? I have concern that since the athlete was only 1 mmol/L under the threshold, this may be a spurious finding. Were there any EAH symptoms? Was this noted during the marathon attempt? At the least, the authors should note the distinction between EAH and slight sodium depletion observed in this study – especially given the title of the paper. Perhaps the authors should soften the title of the paper given the findings. I encourage the authors to review the latest EAH guidelines (https://www.ncbi.nlm.nih.gov/pmc/articles/PMC5334560/).
LINE 135: The relationship between HR and running speed is very well known. I don’t believe this data or section adds anything novel and does not require its own section.
Author Response
Reviewer 1
Knechtle et al present an interesting case-study in an understudied population. While the data presented is certainly informative from a self-pacing point of view, I have some concerns about the interpretation and main findings of the paper.
LINE 51: What were the post-race instructions for the athlete? Was there any specific dietary or activity instructions? The 5-day post-race time point also has a fairly low sodium level, similar to the post-race value.
Answer: We agree with the expert reviewer and added ‘The athlete prepared for this event by recommendations of his parents, his grandparents and his teacher regarding training and equipment. No specific recommendations were followed regarding food and fluid intake before, during and after the event’ in the method section. The sodium concentration on day 5 post-race was > 135 mmol/l.
LINE 56: More details about the preparation are required. Was this self-taught? Was there a coach? Was there specific instructions for fluid intake and pacing in the pre-race recommendations? What was the pre-race preparation (fluids/food)? All these elements could impact both the pacing, and fluid intake findings of the study.
Answer: We agree with the expert reviewer and added ‘The athlete prepared for this event by recommendations of his parents, his grandparents and his teacher regarding training and equipment. No specific recommendations were followed regarding food and fluid intake before, during and after the event’ in the method section.
LINE 64: One of the main results of the study is that fluid intake increased in the second half of the run. Was this confounded by the additional 2 support team members that followed the athlete? Were fluids offered to the athlete, or asked for unprompted? If this is a major finding, more methodological details should be presented.
Answer: We agree with the expert reviewer and added ‘The runner got liquid and food not according to a fixed scheme but according to his needs’.
Some of the statistics seem inappropriate for the case study design. Since this is an n=1 study, I fear there is too much over interpretation of the statistics presented for the data. Presentation of case data with subjective trend analysis is likely adequate for this paper; there isn’t enough to make use of more complicated statistics to interpret the findings for generalizability.
Answer: We agree with the expert reviewer and added this aspect in the methods (“The abovementioned statistics should be considered with caution in terms of interpretation of the findings for generalizability considering the nature of the study (case study, n=1).”).
LINE 79: Why either linear or 4th degree polynomial? Was any F testing done to examine which polynomial gave the best fit?
Answer: We agree with the expert reviewer and added this information in the methods (“- depending on wwhich one polynomial provided the best fit according to F test -”).
LINE 79: Regressions with time-series data (or distance in this case – similar issue), do not give true relationships amongst variables, since the dependent is usually a function of time. The trend line is fine, but there is likely no need to show the confidence bars in the plots.
Answer: We agree with the expert reviewer and removed the HR-speed figure and the confidence intervals in the plots from figures 1 and 2.
LINE 82: How was an ANOVA used when there was no variance at any of the Distance measures? It appears the authors grouped their km data into bins. Why 7km bins? 6 splits of 7km is 35km… where did the rest of the km get split?
Answer: We agree with the expert reviewer and the explanation was the one he mentioned, i.e. since there was no variance across distance, the overall distance was grouped into six 7km splits (6x7km=42km) allowing the use of ANOVA. To make clear the meaning of split, we added in the methods an explanation (“It should be highlighted that split for the purpose of the present analysis referred to distance interval rather than to a specific distance point.”).
FIG1: Fluid intake seems to be measured in an ordinal scale (either 1 or 2 dL units). It would be inappropriate to use any regression slopes with this data.
Answer: We agree with the expert reviewer and removed the regression slopes from fluid and caloric data in Fig1.
FIG4 & 5: Please provide a citation for the reference ranges.
Answer: We agree with the expert reviewer and added (reference values from www.risch.ch/de/ribook) in the legend of the figures
LINE 122: Are the reference values for EAH appropriate for adolescent athletes? I have concern that since the athlete was only 1 mmol/L under the threshold, this may be a spurious finding. Were there any EAH symptoms? Was this noted during the marathon attempt? At the least, the authors should note the distinction between EAH and slight sodium depletion observed in this study – especially given the title of the paper. Perhaps the authors should soften the title of the paper given the findings. I encourage the authors to review the latest EAH guidelines (https://www.ncbi.nlm.nih.gov/pmc/articles/PMC5334560/).
Answer: We agree with the expert reviewer and insert in the section ‘subject and event’ the following ‘The athlete showed no signs of dehydration and/or hyponatremia’. Regarding the title and EAH, the cited paper writes in the Introduction ‘For most labs, the diagnostic threshold for hyponatremia is any blood [Na+] below135 mmol/L regardless of the presence or absence of signs and symptoms’. We therefore make no changes in the title since the plasma sodium was 134 mmol/l after the marathon and therefore fulfilled the requested threshold for EAH.
LINE 135: The relationship between HR and running speed is very well known. I don’t believe this data or section adds anything novel and does not require its own section.
Answer: We agree with the expert reviewer and removed this aspect from all parts of the text. E.g. “Heart rate correlated with running speed (r=0.55, p<0.05).” was deleted from the abstract; “HR correlated largely with speed (Figure 3).” and Figure 3 were deleted from the Results section, and 4.2 section was deleted in the discussion.
Reviewer 2 Report
In this case study, the authors tracked a 15 year old male before, immediately post, and for 5 days following completion of a marathon. Fluid ingestion and pace was tracked during the marathon and blood biomarkers were assessed before and following.
It is not clear why the authors chose the specific biomarkers - especially biomarkers that are consistently found to be elevated following marathons.
The conclusion stating that a teenager was able to run a marathon with no harmful effects on health is too broad of a statement and not accurate. His biomarkers of inflammation and muscle damage were elevated following the marathon, demonstrating an acute negative effect of the marathon, despite values returning to normal after 5 days. The teenager also demonstrated hyponatremia following the run. In addition, there could be potential harmful effects - but there were many variables that were not measured and follow up only lasted for 5 days.
Was the participant's body mass taken before or following the race (and into recovery)? Could weight loss or hydration have an effect on the biomarkers?
Author Response
Reviewer 2
In this case study, the authors tracked a 15 year old male before, immediately post, and for 5 days following completion of a marathon. Fluid ingestion and pace was tracked during the marathon and blood biomarkers were assessed before and following.
It is not clear why the authors chose the specific biomarkers - especially biomarkers that are consistently found to be elevated following marathons.
Answer: We agree with the expert reviewer that these markers were consistently found to be elevated after a marathon. This is the reason that we also determined these markers in order to compare to existing literature.
The conclusion stating that a teenager was able to run a marathon with no harmful effects on health is too broad of a statement and not accurate. His biomarkers of inflammation and muscle damage were elevated following the marathon, demonstrating an acute negative effect of the marathon, despite values returning to normal after 5 days. The teenager also demonstrated hyponatremia following the run. In addition, there could be potential harmful effects - but there were many variables that were not measured and follow up only lasted for 5 days.
Answer: We agree with the expert reviewer and changed in the abstract and in the manuscript to ‘In summary, a male teenager at the age of 15 years was able to run a marathon below 6 hours developing asymptomatic EAH and an increase in leucocytes, CRP, CK and LDH as markers of inflammation and skeletal muscle damage’.
Was the participant's body mass taken before or following the race (and into recovery)? Could weight loss or hydration have an effect on the biomarkers?
Answer: We agree with the expert reviewer and added in the discussion ‘A limitation of this case report is the fact that body mass and urinary parameters (e.g. urine specific gravity, urine osmolality) were not measured to quantify hydration status and signs of dehydration. Weight loss and changes in hydration status might have had an effect on the biomarkers.’